# CLOSED-LOOP CONTROL FOR ONLINE CONTINUAL LEARNING

## ABSTRACT

Online class-incremental continual learning (CL) deals with the sequential task learning problem in a realistic non-stationary setting with a single-pass through of data. Replay-based CL methods have shown promising results in several online class-incremental continual learning benchmarks. However, these replay methods typically assume pre-defined and fixed replay dynamics, which is suboptimal. This paper introduces a closed-loop continual learning framework, which obtains a real-time feedback learning signal via an additional test memory and then adapts the replay dynamics accordingly. More specifically, we propose a reinforcement learning-based method to dynamically adjust replay hyperparameters online to balance the stability and plasticity trade-off in continual learning. To address the non-stationarity in the continual learning environment, we employ a Q function with task-specific and task-shared components to support fast adaptation. The proposed method is applied to improve state-of-the-art replay-based methods and achieves superior performance on popular benchmarks.

## 1 INTRODUCTION

A major challenge in research on artificial neural networks is to develop incremental learning ability to accumulate knowledge over time from a non-stationary stream of data. Most deep learning techniques are designed with the assumption that the training samples are drawn independently and identically from a fixed data distribution, but this assumption is often violated in ever-changing real-world environments. Most successful deep learning results are achieved by training on pre-collected datasets. The resulting model is static and incapable of adapting its behavior to the non-stationary environment over time (Delange et al., 2021). When new data becomes available, the training process needs to restart again. In a non-stationary environment, training on the new data can seriously undermine the model's previously acquired knowledge of past data, which may cause what is known as "catastrophic forgetting" (Parisi et al., 2019). In many real-world applications, a large amount of new data are generated continually, and it has become increasingly important to ensure that the learning agent has lifelong learning capabilities to learn new knowledge and maintain the old knowledge.

Our work focuses on online class incremental (OCI) continual learning problems to learn a sequence of classification tasks with a single pass over the data. To address the problem of catastrophic forgetting in this setting, continual learning algorithms need to balance between stability and plasticity: the ability to robustly maintain old knowledge while being able to quickly incorporate new knowledge. Previous work on continual learning deals with this problem by modifying the standard training process using memory replay-based, regularization-based, or parameter isolation-based techniques. The work presented in this paper rests on the observation that, besides requiring changes to the training algorithm, the sequential learning process also brings special challenges to the hyperparameter tuning process, which has not been extensively studied so far. In fact, a large number of CL methods have to rely on well-selected hyperparameters to effectively balance the trade-off between stability and plasticity; to search for the best hyperparameters, most research in this area follows an offline approach to go through the whole data sequence several times measuring performance against a representative, held-out validation set. This violates the fundamental assumption of continual learning, namely that there is no access to previous task data. There is some work on heuristic hyperparameter tuning specifically designed for online CL (Aljundi et al., 2019; Mai et al., 2021a), but it requires extra held-out task validation data, which may not be available.

To address the hyperparameter tuning problem, we propose a closed-loop continual learning framework that is able to balance stability and plasticity without access to an external held-out validation set. Specifically, we build a real-time feedback signal that measures the performance of the CL agent and enables online hyperparameter tuning using reinforcement learning (RL). The proposed closed-loop CL framework is combined with replay-based continual learning to achieve adaptive memory replay for online class-incremental continual learning.

Replay-based CL methods prevent forgetting of previously learned knowledge by maintaining a small memory of past data and performing joint training on incoming and memorized data during the optimization process. Despite their simplicity, these methods have significantly outperformed CL approaches without episodic memory in the online class-incremental setting (Chaudhry et al., 2019). Compared to standard supervised learning, two main challenges exist in memory replay methods. One is the risk of overfitting the memory, as a relatively small amount of memorized samples are repeatedly rehearsed in the training process. The other challenge is the class imbalance problem in the joint training process: generally, only a relatively small amount of old-class data is available in the memory. Because of these challenges, it is important to be able to modulate the replay dynamics of these methods by hyperparameters such as the replay step size and the number of replay iterations. However, the values assigned to these hyperparameters are often hand-crafted and kept fixed during the whole learning process in previous research on memory replay methods. In this work, we address this problem by proposing a method that is able to dynamically adapt the hyperparameters of memory replay methods in an online manner. Specifically, based on a closed-loop CL framework, we apply reinforcement learning to adjust replay hyperparameters to achieve a dynamic adjustment between stability and plasticity. We formulate the problem of adjusting replay hyperparameters as a Markov decision process and use a purpose-built state space to capture the current performance of the CL model and the relevant input data information for replay hyperparameter decision-making. To address the non-stationarity of the environment, we propose a Q function with task-shared and task-specific components. The proposed RL-based methods are used to improve the two replay-based methods ER (Chaudhry et al., 2019) and SCR (Mai et al., 2021b), and achieve adaptive memory replay.

## 2 Related Work

### 2.1 Continual Learning

Continual learning is concerned with learning a sequence of tasks. In each task $t$, data $\mathcal{X}^{(t)}$ is sampled from distribution $D(t)$; the corresponding class labels are $\mathcal{Y}^{(t)}$. The goal is to minimize the loss on all seen tasks given limited or no access to data $\mathcal{X}^{(t)}, \mathcal{Y}^{(t)}$ from previous tasks $t < T$: $\sum_{t=1}^{\mathcal{T}} \mathbb{E}_{\left(\mathcal{X}^{(t)}, \mathcal{Y}^{(t)}\right)} \left[\ell\left(f\left(\mathcal{X}^{(t)}; \theta\right), \mathcal{Y}^{(t)}\right)\right]$ with loss function $\ell$, CL parameters $\theta$, $\mathcal{T}$ the number of tasks seen so far, and $f$ representing the CL network function. In recent years, continual learning methods have been proposed to deal with different problem settings and evaluation protocols. Van de Ven & Tolias (2019) summarize three continual learning scenarios based on whether task identity is provided at test time. Notably, there are substantial differences between the difficulties of these three scenarios. The first scenario is task-incremental, which assumes the task identity is provided at test time. Methods proposed for this scenario usually train task-specific components to prevent the interference between new tasks and old task (Yoon et al., 2020; 2018; Mallya & Lazebnik, 2018). A common practice in this setting is to use multi-head evaluation with a separate output layer (head) assigned to each task. The task ID is used to determine the head to use for inference. This scenario is the easiest since the model just needs to distinguish classes within a task. The second scenario is domain incremental, where the structure of the task is always the same and the input distribution is changing. The last scenario is class-incremental, with new tasks containing new classes. This setting employs single-head evaluation, meaning the CL agent needs to classify all the classes seen so far, without known task information. This setting can be used as a realistic framework of the common real-world problem of incrementally learning new classes of objects.

Another aspect of CL is whether offline or online learning is considered. The offline CL setting assumes full access to the whole data for a task at once. Therefore, training on each single task can be performed over multiple epochs, before moving on to the next task (Rebuffi et al., 2017).

The online CL setting is more challenging: a stream of samples is seen only once and the sampling distribution is non-IID. In this paper, we focus on this more challenging and realistic setting.

To combat catastrophic forgetting, CL methods can be categorized into three families: replay methods, regularization-based methods, and parameter isolation methods (Delange et al., 2021; Parisi et al., 2019). The regularization methods, such as EWC++ (Chaudhry et al., 2018a) and LWF (Li & Hoiem, 2017) introduce a regularization term in the loss function to prevent dramatic changes to important parameters and consolidate previous knowledge. Parameter isolation-based methods assign different subsets of model parameters to different tasks to prevent the interference between tasks. Some examples include methods that dynamically expand the network, such as RCL (Xu & Zhu, 2018), APD (Yoon et al., 2020), and DEN (Yoon et al., 2018); alternatively, one can use a fixed network (Mallya & Lazebnik, 2018). This line of work generally requires knowledge of the task ID at test time and thus is applied in the task-incremental scenario. Replay-based methods retain a memory to store past data for joint training(Chaudhry et al., 2019; Prabhu et al., 2020). Some methods in this area focus on how to store and sample informative memory samples, such as MIR (Aljundi et al., 2019). Recent work uses supervised contrastive loss in memory-replay to help learn a high-quality feature representation (Mai et al., 2021b). Some work also combines the regularization and replay approaches, such as MC-SGD (Mirzadeh et al., 2020), ICARL (Rebuffi et al., 2017).

In terms of hyperparameter tuning for continual learning, most work in CL simply employs offline hyperparameter optimization using a held-out validation set. However, there is some recent work that discusses hyperparameter tuning frameworks specifically designed for continual learning settings without a representative validation set. For offline continual learning, Delange et al. (2021) proposes a method to dynamically balance the stability-plasticity trade-off via a maximal plasticity search followed by stability decay. Instead of requiring a validation set covering all the tasks, this method makes use of a held-out validation set for a new task; however, some parameters, like the tolerance threshold and decaying speed, need to be manually set in advance. Considering the online continual learning setting, Chaudhry et al. (2018b) and Mai et al. (2021a) employ a hyperparameter tuning protocol targeting this setting. These approaches use external cross-validation data streaming with a small number of tasks. Offline hyperparameter tuning is applied on this external validation data with multiple passes to identify optimal values, which are then used for the actual online continual learning tasks. A limitation of this work is that it relies on external validation data. Depending on the similarity between the validation tasks and the actual continual learning tasks, the chosen hyperparameters may not be appropriate.

## 2.2 REINFORCEMENT LEARNING

Reinforcement learning (RL) concerns the problem of sequential decision-making in a stochastic environment (Sutton & Barto, 2018). In standard reinforcement learning settings, an RL agent interacts with the environment to maximize the long-term rewards. Typically, this interactive process is formulated as a Markov decision process (MDP) described by a tuple $< S, A, R, \mathcal{P}, \gamma >$, where $S$ is the state space, $A$ is the action space, $R : S \times A \to R$ is the reward function, $\mathcal{P} : S \times A \times A \to [0, 1]$ is the state transition probability, and $\gamma$ is the discount factor. The agent behavior is defined as the policy $\pi : S \to A$ that determines how and action is selected during the interaction. The goal of the RL agent is to learn a policy that maximizes the cumulative rewards in an episode. The action-value function $Q_\pi(s, a)$ is used to represent the expected long-term reward of executing action $a$ in state $s$, where $s_0$ is the initial state distribution.

Although reinforcement learning has achieved superior performance in many challenging applications such as board games, continual locomotion control, and complex computer games (Schrittwieser et al., 2020; Mnih et al., 2015; Vinyals et al., 2019), its application to adapting the learning process of other machine learning algorithms is relatively limited. Reinforcement learning and bandits algorithms have been used in the area of neural network architecture search and offline hyperparameter optimization in IID environments where a held-out validation set is available and the reward distribution is static (Li et al., 2017; Parker-Holder et al., 2020; Hoffman et al., 2014). To our knowledge, the only work applying reinforcement learning in continual learning is in the parameter isolation-based method, which applies it to dynamically expand a network (Xu & Zhu, 2018). This approach is applied in the offline task-incremental continual learning setting and assumes access to a held-out validation set. There does not appear to be any work that applies reinforcement learning to replay-based continual learning methods or online continual learning settings.

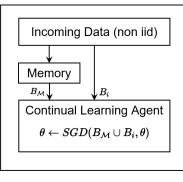 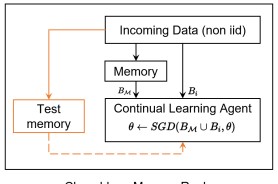

Figure 1: Closed-loop continual learning framework: an additional test memory is constructed to approximate validation data and used as feedback signal to guide online hyperparameter adaptation.

## 3 CLOSED-LOOP CONTINUAL LEARNING

To tune the hyperparameters of continual learning for a never-ending data stream, we propose a closed-loop continual learning framework, as shown in Figure 1. The main idea is to design a quantitative feedback signal to measure the performance of a CL agent in an online manner and then use this feedback signal to guide the hyperparameter selection. To this end, we propose to build a test memory online that includes a subset of samples of all the classes seen so far and use the performance of the model on this test memory to approximate its performance on the test data. Evaluating the CL model using the test memory leads to an online feedback signal, which can be used to adjust the hyperparameters of the CL model and construct a closed-loop control system. To construct such test memory, we follow a procedure similar to the one employed in the management of replay memory: Given an incoming batch of data, non-overlapping subsets of samples are stored in the replay memory and test memory respectively. To make the test memory representative, we employ a class-balanced greedy sampling method (Prabhu et al., 2020) to greedily store samples from the data stream with the constraint to asymptotically balance the class distribution. Specifically, when data from a new class arrives, the sampler simply creates a new bucket for that class and starts deleting samples from the class that currently contains the maximum number of samples. Every sample in this class is assumed to be equally important and is randomly chosen for removal.

Notably, a significant difference between the replay and test memory is that the samples in the replay memory will be repeatedly drawn to update model parameters via gradient descent, while the test memory will not take part in the direct update of model parameters, but only be used to evaluate the model's performance and indirectly affect the hyperparameters in the model update.

To investigate the effectiveness of test memory, we examine the correlation between the model's accuracy on the test data and the model's accuracy on the test memory. Our goal is to design a test memory that is representative and correlates well with test data. Meanwhile, since the use of extra test memory leads to a space overhead, we are particularly concerned with whether the test memory can perform well with a tiny test memory size.

## 4 REINFORCEMENT LEARNING-BASED ADAPTIVE REPLAY

We follow the aforementioned closed-loop continual learning framework to treat the test memory performance as a feedback signal and use reinforcement learning as the controller to adjust replay hyperparameters accordingly, as shown in Figure 2. We apply the closed-loop CL framework to balance the stability and plasticity trade-off by dynamically adapting the replay dynamics in terms of the replay ratio and the number of replay iterations. In this section, we firstly describe the overall RL-based adaptive memory replay framework in Section 4.1 and then discuss the design of the Markov decision process to dynamically adjust the replay ratio and the replay iterations in Section 4.2 and Section 4.3 respectively. Lastly, Section 4.4 presents the reinforcement learning algorithm to solve the sequential decision-making problem in the non-stationary continual learning environment.

### 4.1 RL-BASED ADAPTIVE MEMORY REPLAY FRAMEWORK

The key element in replay-based continual learning algorithms is the joint training on memory data of old classes and incoming data of new classes to mitigate catastrophic forgetting. The main challenge is data imbalance between the new classes and old classes in the joint training, with only a

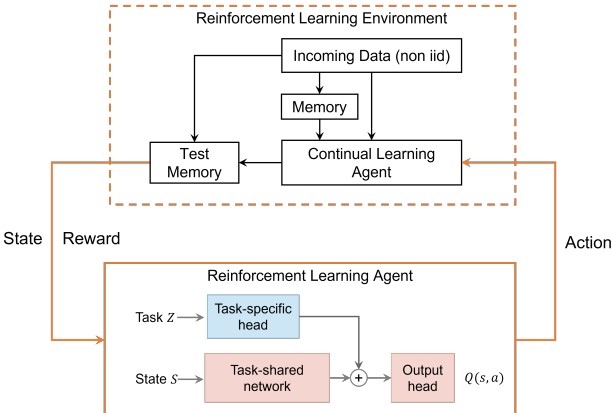

Figure 2: Reinforcement learning-based adaptive replay framework: the RL agent selects replay hyperparameters and the CL agent executes the replay update and evaluates the model on test memory to output reward and state for the RL agent.

small number of old classes data stored in the memory. This class imbalance leads to a bias towards the new classes in the final output layers (Mai et al., 2021a; Wu et al., 2019). Another challenge is the possible overfitting of memory data as a small amount of memory data needs to be replayed multiple times (Verwimp et al., 2021). Ad-hoc replay dynamics have been designed to improve the performance of the replay method. For example, a Review Trick (Mai et al., 2020) is applied in the winning solution of the continual learning challenge at CVPR2020, which employs a small learning rate on the memory at the end of learning each task. And Verwimp et al. (2021) find that isolated training on the memory may be used as a way to move away from the high-loss ridge and reduce the overfitting of the replay memory. Yet other research (Aljundi et al., 2019; Bang et al., 2021) considers how to effectively store or select a representative subset of memory samples for joint training. In contrast, our work focuses on the hyper-parameters of internal replay dynamics, including replay step size and the number of replay iterations.

Table 1 presents the process of dynamically adjusting a replay hyperparameter with reinforcement learning, based on treating the replay ratio as the hyperparameter of concern. At each joint training step, the RL agent analyzes the current state of the environment and selects a hyperparameter value based on the current state. Then, the CL agent executes the joint training step following the specified hyperparameter value. Lastly, the CL model is evaluated on the test memory to generate reward feedback representing the goodness of the hyperparameter choice for the RL agent. To instantiate this algorithm for adaptive memory replay, we need to design the state, reward, and action space. We treat the adaptation of replay ratio and the number of replay iterations as two separate problems as the information used to tune the two hyperparameters is different. The MDP design details are discussed in the following two sections.

## 4.2 ADAPTIVE MEMORY REPLAY RATIO

To formulate the problem of adjusting the replay ratio as a sequential decision-making process, we define the *Action* space as the set of possible values of the replay ratio—the ratio between the stepsize used for training on the incoming batch and the stepsize used for training on the memory batch. As shown in Equation 1, the step size on the memory batch is set to one, and we use $\alpha$ to denote the step size on the incoming batch, which is equal to the replay ratio.

$$J_{replay\_ratio} = \sum_{x \in B_{\mathcal{M}}} \ell(f(x;\theta), y) + \alpha \times \sum_{x \in B_i} \ell(f(x;\theta), y) \tag{1}$$

With a higher replay ratio, the learning process is more focused on the incoming batch and the plasticity of the model is enhanced. Conversely, with a lower replay ratio, the learning process is more focused on the memory samples and the stability of the CL agent is increased. The replay

Table 1: RL-based memory replay hyperparameter adaptation

| **Algorithm: Memory Replay with adaptive replay hyperparameter** |
|---|
| **Input**: $\mathcal{M}$ train memory, $\mathcal{M}_{test}$ test memory, |
| $B_i$ incoming batch data, |
| $\theta$ parameter of CL network, $\phi$ parameter of RL agent, |
| $\alpha$ replay ratio |
| **function** RL_Replay($\mathcal{M}$,$\mathcal{M}_{test}$, $B_i$ ) |
| $\quad B_{\mathcal{M}} \sim \mathcal{M} \triangleright$ Sample memory batch from train memory |
| $\quad s \leftarrow RL\_state(\theta, \mathcal{M}_{test}, B_i, B_{\mathcal{M}}) \triangleright$ Compute state |
| $\quad a \leftarrow \phi(a\|s) \triangleright$ Sample RL action (i.e. replay hyperparameter) |
| $\quad \theta \leftarrow joint\_training(B_{\mathcal{M}} \cup B_i, \theta, a) \triangleright$ Update CL model |
| $\quad r \leftarrow RL\_Reward(\theta, \mathcal{M}_{test}) \triangleright$ Compute Reward |
| $\quad \phi \leftarrow UpdateRL(a, r, s) \triangleright$ Update RL agent |
| $\quad \mathcal{M} \leftarrow B_i \triangleright$ Update train memory |
| $\quad \mathcal{M}_{test} \leftarrow B_i \triangleright$ Update test memory |

ratio is typically set to 1 in previous CL methods. The goal of reinforcement learning is to adapt the replay ratio for each incoming batch to maximize the performance on the test memory.

Our goal is to learn to make a data-dependent replay ratio decision. The intuition is that if the current incoming batch is challenging and important for the CL agent, a high ratio should be employed. An incoming batch is considered being informative and important for the training of the CL agent if it contains some new classes or classes that the CL agent has not learned to recognize well. Following this idea, we design the *State* space to capture the relevant information to measure the importance of the replay batch in order to support the selection of replay ratio. To measure the importance of a replay batch, we compute a weighted model accuracy based on the class distribution of the replay batch and the current class-specific accuracy on the test memory. Specifically, we first compute the current model's accuracy on the test memory for each class. We use $\mathcal{C}$ to denote the classes that have been seen by test memory and $A(c), c \in \mathcal{C}$ is the current prediction accuracy for each class measured using test memory. We use $B_i^{\mathcal{C}} = [x, y \in B_i | y \in \mathcal{C}]$ to denote the subset of incoming batch excluding the samples from the classes that have not been seen by the test memory before. The weighted accuracy of incoming batch is defined as $A_{B_i} = 1/|B_i^{\mathcal{C}}| \sum_{x,y \in B_i^{\mathcal{C}}} A(y)$. If a batch of data contains a lot of classes for which the CL model currently performs poorly (as indicated by the test memory), the weighted accuracy is expected to be low. The weighted accuracy of memory batch $A_{B_{\mathcal{M}}}$ is defined in a similar way. This information is included in the state space to help adjust the step size for this batch. The overall state space includes the current performance of the CL agent measured by the average loss and accuracy using the test memory, and the weighted accuracy/loss and the number of new classes for the incoming batch and memory batch. A detailed description is shown in the appendix (Table 3).

Following the closed-loop CL framework, we use the test memory to construct a reward signal. *Reward* is defined as the decrease of the loss on the test memory after one gradient update step with memory replay:

$$R_t = \sum_{x \in \mathcal{M}_{test}^t} \ell(f(x; \theta^t), y) - \sum_{x \in \mathcal{M}_{test}^{t+1}} \ell(f(x; \theta^{t+1}), y)) \tag{2}$$

where $\theta^{t+1} = \theta^t - \nabla_\theta(\sum_{x \in B_{\mathcal{M}}^t} \ell(f(x; \theta^t), y) + \alpha \times \sum_{x \in B_i^t} \ell(f(x; \theta^t), y))$.

The reward is affected by several factors, including the CL network networks parameters $\theta$, the sampled memory batch $B_{\mathcal{M}}$, the incoming batch $B_i$, and the test memory $\mathcal{M}_{test}$. In Section 4.4, we discuss how to address the non-stationarity of the reward function. To simplify the problem, we employ a one-step MDP to maximize the reward for each memory replay step.

### 4.3 ADAPTIVE MEMORY REPLAY ITERATION

We formulate online tuning of the number of replay iterations as a reinforcement learning problem by defining the *Action* space as the number of additional memory iteration steps for each incoming

batch. For each incoming batch, the CL agent first performs a vanilla memory replay step. After that, the RL agent decides how many more memory replay iterations will be conducted for this incoming batch. During each memory iteration step, the CL agent repeats the process of replaying memory, i.e., sampling a memory batch and concatenating it with the incoming batch to perform gradient updates.

Previous work has been largely restricted to performing only a single memory iteration on incoming samples. For online continual learning, multiple memory iterations can be particularly important, as this can help maximally extract information from a single pass through the incoming data samples. Nevertheless, with more iterations, the risk of overfitting the memory data increases and can hurt the performance in some cases. To support the selection of the number of memory iterations, we again use a state space design that captures relevant information, including the loss and accuracy on the test memory batch, and the training loss and accuracy on the incoming batch. Similar to the replay ratio adaptation, *Reward* is defined as the decrease of CL loss measured on the test memory (see Equation 2), and we employ a one-step MDP to maximize the reward for each memory replay step.

### 4.4 REINFORCEMENT LEARNING ALGORITHM

To solve the two sequential decision-making problems designed above, DQN (Van Hasselt et al., 2016) is used to update the RL agent, and the Q function $Q(s, q)$ is learned with experience replay techniques. We use a Q function to model the relationship between the choice of replay hyperparameters and the resulting model performance improvement measured on the test memory.

It is worth mentioning that the reinforcement learning environment in the closed-loop continual learning framework contains several important components: incoming data, memory, test memory, and CL agent (see Figure 2). The reward function is influenced by these factors as shown in Equation 2. In the continual learning environment, the distribution of incoming data is changing from task to task. As the training continues, the state of the CL agent is also changing as it moves to the local minimum. Due to these factors, the influence of the replay hyperparameters on the model performance may change dynamically during different continual learning stages. Therefore, it is reasonable to assume the Q function may not be totally the same for different tasks, although it does share some common properties. To address this non-stationarity of the environment, we propose a task-dependent Q learning structure.

As shown in the RL agent part of Figure 2, when a new CL task arrives, instead of training a completely new Q function or totally copying the old Q function from previous tasks, we propose a Q network with a task-shared component and task-specific component. The task-shared component is used to capture the shared closed-loop control information across tasks and the task-specific component is randomly initialized for each new task. It should be noted that although the goal of CL is to perform well on all the tasks seen before, the goal of RL is to estimate an accurate Q function for the current task, regardless of previous tasks. This is because the Q function is used to adapt and improve the training process of the current CL task. The process of learning the Q function is similar to transfer learning. There is no need to store the task-specific head for the previous tasks. Also, although there is a task-specific component in the RL network, there is no task-specific component in the CL network. Thus, unlike in the expansion-based CL methods, no task information is needed at inference time.

## 5 EXPERIMENTS

In our evaluations, we focus on the comparison of the RL-based method to random sampling in the experience replay (ER) (Chaudhry et al., 2019) and SCR (Mai et al., 2021b) approaches, which our method directly modifies. ER employs vanilla memory replay with a replay ratio of 1 and 1 memory iteration. Supervised Contrastive Replay (SCR) applies supervised contrastive loss, instead of cross-entropy loss, for the joint training and improves the performance of vanilla replay by a large margin.[1] We also consider the following reference baselines:

---

[1]Note that SCR employs Nearest-Class Mean classifier which is expensive for online learning and testing since the feature space is changing during CL training and the class centers need to be recomputed repeatedly. To employ SCR in a closed-loop CL setting, we add a softmax head for SCR.

- finetune: Trains continuously on new tasks without any forgetting avoidance strategy.
- MIR (Aljundi et al., 2019): Maximally Interfered Retrieval, another replay-based method that improves ER by using a controlled sampling of memories for replay.
- AGEM (Chaudhry et al., 2018b): Averaged Gradient Episodic Memory, which utilizes the samples in the memory to constrain the CL parameter updates instead of using direct replay.
- LWF (Li & Hoiem, 2017):LearningWithout Forgetting, a regularization method that utilizes knowledge distillation to penalize feature drift.

The experiments are conducted using three continual learning benchmarks. Implementation details can be found in the appendix.

- CIFAR-100-Split divides CIFAR-100 dataset into 20 disjoint tasks. Each task contains 5 classes. There are 500 images for training and 100 images for testing in each class.
- CIFAR-10-split splits CIFAR-10 dataset into 5 disjoint tasks. Each task contains 2 classes. There are 5000 images for training and 1000 images for testing in each class.
- CLRS (Li et al., 2020) is a continual learning benchmark for remote sensing image scene classification. It contains 25 scene classes, split into 5 disjoint tasks. Each class has 450 training data and 150 test data.

## 5.1 CLOSED-LOOP CONTINUAL LEARNING

We first examine the effectiveness of the test memory. We compare the loss on the test memory with the loss on the test data. We use training memory loss as a baseline for approximating the test loss. As shown in the Figure 3, the loss computed on test memory is closely related to loss on test data. Notably, the loss computed on training memory is quite different from the loss on test data, especially when more continual learning tasks are encountered. This result shows the introduce of additional test memory is necessary.

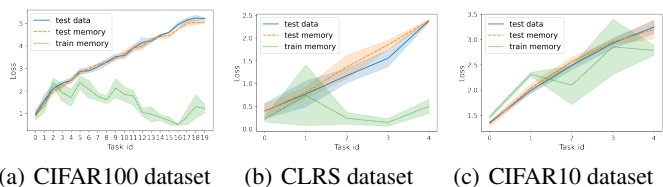

(a) CIFAR100 dataset    (b) CLRS dataset    (c) CIFAR10 dataset

Figure 3: Approximation of test loss using test memory and training memory.

To study whether a small test memory is effective, we further investigate the influence of test memory size on the approximation quality of test loss. Specifically, we analyze the correlation between the model's performance on the test memory and on the test data. In all three datasets, the correlation between the test data and test memory is above 0.9 when the test memory size is above 100, as shown in Figure 5. This result suggests that a small test memory is enough to present reliable feedback on the model's performance.

## 5.2 ADAPTIVE REPLAY RATIO

We now consider the performance of our RL-based methods relative to the baselines. The results in Table 2 shows that our approach can improve accuracy for both ER and SCR. The replay ratio chosen by the ER-RL agent as the greedy action in each task is shown in Figure 4(a). In the first few tasks, the RL agent explores all the choices randomly and then slowly focuses on more desirable ratio values. In this case, we can see that RL favors a smaller step size on the incoming batch. As mentioned earlier, one challenge in the replay-based method is the bias toward new classes due to class imbalance in the joint training. With a small replay ratio, the joint training process gives a higher weight on the old classes and can help to reduce the bias towards new classes and mitigate catastrophic forgetting. However, it should be noted that although the agent is mostly focused on smaller values of incoming ratio, large values are still considered to be the greedy action once in

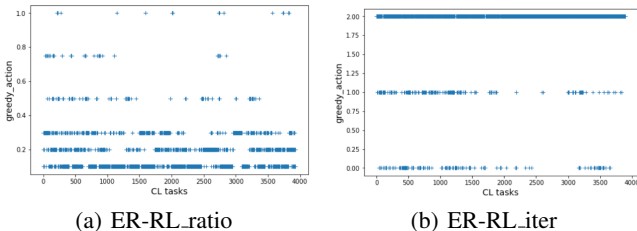

(a) ER-RL_ratio        (b) ER-RL_iter

Figure 4: RL-based replay hyperparameter adaptation (CIFAR100): a) replay ratio selection percentage in each task; b) replay iteration selection percentage in each task.

while by the RL agent. The selection of large incoming ratio once in a while is likely beneficial to combat outfitting on replay memory.

Table 2: Continual learning performance in three datasets across 3 runs.

| Methods | CIFAR100 | CLRS | CIFAR10 |
|---|---|---|---|
| finetune | 2.5 ($\pm$0.4) | 4.4 ($\pm$0.3) | 13.4 ($\pm$1.5) |
| LWF | 8.8 ($\pm$0.3) | 9.2 ($\pm$2.2) | 18.5 ($\pm$1.3) |
| AGEM | 3.2 ($\pm$0.3) | 6.4 ($\pm$0.6) | 17.2 ($\pm$1.2) |
| MIR | 21.5 ($\pm$1.5) | 12.4 ($\pm$1.3) | 57.2 ($\pm$5.4) |
| ER | 20.8 ($\pm$1.5) | 15.2 ($\pm$1.1) | 54.1 ($\pm$3.3) |
| ER-RL_ratio (ours) | 22.9 ($\pm$1.8) | 16.8 ($\pm$0.8) | 59.2 ($\pm$3.3) |
| SCR | 38.4 ($\pm$0.8) | 24.1 ($\pm$3.5) | 70.2 ($\pm$3.0) |
| SCR-RL_ratio (ours) | **39.3 ($\pm$0.5)** | **26.0 ($\pm$2.1)** | **73.3 ($\pm$1.1)** |
| **Multiple Iterations** | | | |
| ER-3iter | 27.9 ($\pm$0.7) | 17.8 ($\pm$1.8) | 66.7 ($\pm$1.8) |
| RL_ER_iter (ours) | 28.1 ($\pm$0.7) | 17.0 ($\pm$0.7) | 68.0 ($\pm$2.2) |
| SCR-3iter | **46.6 ($\pm$0.5)** | 30.7 ($\pm$6.0) | **78.2 ($\pm$0.5)** |
| RL_SCR_iter (ours) | 44.2 ($\pm$0.1) | **32.0 ($\pm$2.1)** | 77.0 ($\pm$1.2) |

## 5.3 ADAPTIVE NUMBER OF REPLAY ITERATIONS

Previous replay methods usually employ memory replay with only one iteration for online continual learning. Our results in Table 2 show that multiple iterations can improve results by a large margin for both ER and SCR. Thus, the number of memory iterations should be regarded as an important hyperparameter in online continual learning. Table 2 also shows that the reinforcement learning-based method can identify a suitable number of memory iterations in an online manner and achieve performance close to the best possible performance (which is attained by using a fixed number of 3 iterations throughout). Figure 4(b) shows the number of memory iterations selected by the ER-RL agent as the greedy action in each task. We can see that for the first few tasks, the RL agent explores the environment and selects all the choices with nearly equal probability. As the learning process continues, the RL agent gradually focuses on the more promising number of memory iterations (note that the figure shows the number of *additional* iterations selected).

## 6 CONCLUSION

This paper proposes a closed-loop continual learning framework to adjust replay dynamics in an online manner. We provide empirical evidence to show that a tiny test memory can provide a reliable feedback signal with a strong correlation to ground truth and can be used to guide the learning process of CL agents. We present a concrete example of employing closed-loop continual learning to achieve adaptive memory replay for online class-incremental continual learning. Reinforcement learning methods are applied to autonomously adjust the replay ratio and the number of replay

iterations and dynamically balance the plasticity and stability trade-off. Future work will investigate the optimization of other hyperparameters such as the learning rate and regularization parameters.

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

## A  APPENDIX

### A.1  MDP DESIGN

The MDP design for replay ratio adaptation is shown in Table 3. The MDP design for adaptation of the number of replay iterations is shown in Table 4.

Table 3: MDP design for RL-based replay ratio adaptation

| |
| --- |
| **State:** |
| average loss and accuracy of test memory $\frac{1}{|B_{\mathcal{M}_{test}}|} \sum_{x \in B_{\mathcal{M}_{test}}} \ell(f(x;\theta), y)$ |
| weighted loss and accuracy of incoming batch $A_{B_i}$ |
| weighted loss and accuracy of memory batch $A_{B_{\mathcal{M}}}$ |
| number of unseen classes in the incoming batch $|B_i/B_i^{\mathcal{C}}|$ |
| number of unseen classes in the memory batch $|B_{\mathcal{M}}/B_{\mathcal{M}}^{\mathcal{C}}|$ |
| **Action**: replay ratio $[0.1, 0.2, 0.3, 0.5, 0.75, 1.0, 1.2, 1.5]$ |
| **Reward**: decrease in the test memory loss |

Table 4: MDP design for RL-based replay iteration adaptation

| |
| --- |
| **State:** |
| average loss and accuracy of test memory $\frac{1}{|B_{\mathcal{M}_{test}}|} \sum_{x \in B_{\mathcal{M}_{test}}} \ell(f(x;\theta), y)$ |
| train loss and accuracy of incoming batch $\frac{1}{|B_i|} \sum_{x \in B_i} \ell(f(x;\theta), y)$ |
| **Action**: additional replay iteration $[0, 1, 2]$ |
| **Reward**: decrease in the test memory loss |

## A.2    IMPLEMENTATION DETAILS

Following (Chaudhry et al., 2019; Mai et al., 2021b), a reduced ResNet18 is used as the backbone model for all datasets. We use stochastic gradient descent with a learning rate of 0.1, and the model receives a batch with size 10 at a time from the data stream. All the methods are trained with cross-entropy loss except that SCR is trained with supervised contrastive loss. The softmax classifier is used for classification. We use a softmax head for SCR which includes a hidden layer of 1024. We use reservoir sampling (Vitter, 1985) for memory management and greedy sampling (Prabhu et al., 2020). We use a memory batch size 10, except that SCR uses a memory batch size of 100. All the experiments are run across three seeds.

In the reinforcement learning, the shared component of the Q network is a 3-layer MLP with size 32. The task-specific component and the output head is a linear layer with size 10. Replay buffer size is 100. Q learning batch size is 50. The learning rate for updating the RL agent is 0.001. The weight decay factor is 0.0001. $\epsilon$-greedy is used for exploration. The exploration rate is reduced from 1 to 0.2 at 20% time steps, then reduced from 0.2 to 0.05 at 50% time steps, and fixed at 0.02 after that.

## A.3    ALGORITHM

## A.4    SUPPLEMENTARY RESULTS

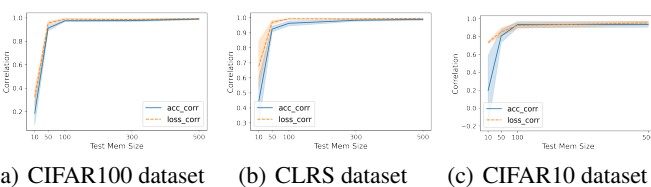

(a) CIFAR100 dataset    (b) CLRS dataset    (c) CIFAR10 dataset

Figure 5: Quality of feedback signal with respect to test buffer size in terms of correlation with real test loss.

Table 5: RL-based memory replay with adaptive replay ratio

| **Algorithm: Joint training with adaptive replay ratio** |
| --- |
| **Input**: $\mathcal{M}$ train memory, $\mathcal{M}_{test}$ test memory, |
| $B_i$ incoming batch data, |
| $\theta$ parameter of CL network, $\phi$ parameter of RL agent, |
| $\alpha$ replay ratio |
| **function** RL_Replay_ratio($\mathcal{M}$,$\mathcal{M}_{test}$, $B_i$ ) |
| $\quad B_\mathcal{M} \sim \mathcal{M} \triangleright$ Sample memory batch from train memory |
| $\quad s \leftarrow RL\_state(\theta, \mathcal{M}_{test}, B_i, B_\mathcal{M}) \triangleright$ Compute state |
| $\quad a \leftarrow \phi(a|s) \triangleright$ Sample RL action |
| $\quad \alpha \leftarrow Set\_Replay\_Para(a) \triangleright$ Set replay memory iterations |
| $\quad \theta \leftarrow SGD(B_\mathcal{M} \cup B_i, \theta, \alpha) \triangleright$ Update CL model |
| $\quad r \leftarrow RL\_Reward(\theta, \mathcal{M}_{test}) \triangleright$ Compute Reward |
| $\quad \phi \leftarrow UpdateRL(a, r, s) \triangleright$ Update RL agent |
| $\quad \mathcal{M} \leftarrow B_i \triangleright$ Update train memory |
| $\quad \mathcal{M}_{test} \leftarrow B_i \triangleright$ Update test memory |

Table 6: Memory replay with adaptive replay iteration

| **Algorithm: Joint training with adaptive replay iteration** |
| --- |
| **Input**: $\mathcal{M}$ train memory, $\mathcal{M}_{test}$ test memory, |
| $B_i$ incoming batch data, |
| $\theta$ parameter of CL network, $\phi$ parameter of RL agent, |
| $k$ additional replay iteration |
| **function** RL_Replay_iter($\mathcal{M}$,$\mathcal{M}_{test}$, $B_i$ ) |
| $\quad B_m \sim \mathcal{M} \triangleright$ Sample memory batch from train memory |
| $\quad \theta \leftarrow SGD(B_m \cup B_i, \theta) \triangleright$ Update CL model |
| $\quad s \leftarrow RL\_state(\theta, \mathcal{M}_{test}, B_i, B_\mathcal{M}) \triangleright$ Compute state |
| $\quad a \leftarrow \phi(a|s) \triangleright$ Sample RL action |
| $\quad k \leftarrow Set\_Replay\_Para(a) \triangleright$ Set replay memory iterations |
| $\quad$ for i in range(k): |
| $\quad\quad B_m \sim \mathcal{M} \triangleright$ Sample memory batch from train memory |
| $\quad\quad \theta \leftarrow SGD(B_m \cup B_i, \theta) \triangleright$ Update CL model |
| $\quad r \leftarrow RL\_Reward(\theta, \mathcal{M}_{test}) \triangleright$ Compute Reward |
| $\quad \phi \leftarrow UpdateRL(a, r, s) \triangleright$ Update RL agent |
| $\quad \mathcal{M} \leftarrow B_i \triangleright$ Update train memory |
| $\quad \mathcal{M}_{test} \leftarrow B_i \triangleright$ Update test memory |

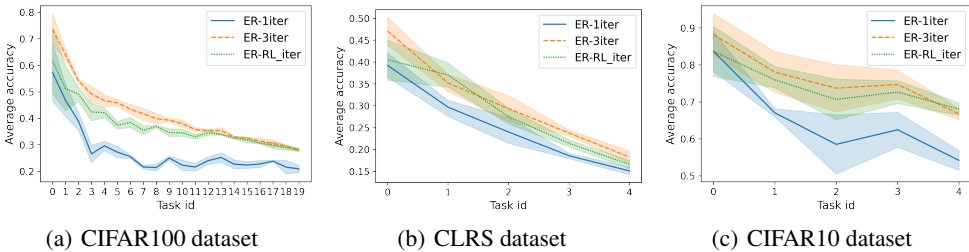

(a) CIFAR100 dataset     (b) CLRS dataset     (c) CIFAR10 dataset

Figure 6: Adaptive memory iteration.

