# OpenReview forum: "Closed-loop Control for Online Continual Learning"
_ICLR.cc/2022/Conference — ICLR 2022 Submitted_

### Official Review · Reviewer_hbHU · 2021-10-31

**Correctness:** 3
**Technical Novelty And Significance:** 3
**Empirical Novelty And Significance:** 3
**Recommendation:** 6
**Confidence:** 4

**Main Review:**

Strengths:
The main strength of the paper is addressing an important problem in continual learning: how to balance between online and memory-based updates, and how many times to iterate when a mini-batch is drawn from the memory.

The paper is well written and mostly understandable. There are many variants of continual learning tasks, among which this paper focuses on an important kind, where a stream of samples is seen only once, and new tasks may contain new classes but no known task information.

Another strength of the work is the simplicity of the idea, which is as simple as using reinforcement learning to learn the hyperparameters. The ingenuity is the employment of a test memory to structure this learning.

Weaknesses:

The main weakness of the work is the lack of clarity of the MDP formulation. In order to formulate a problem with an MDP, there need to be meaningful state-to-state transitions, and the optimal policy needs to be a function of the state. To understand whether those are present in the adopted continual learning setup, one could take a simplified scenario where the overall process of the continual learning setup is stationary. In that scenario, could you give a sensible description of the state-to-state transition dynamics that is contingent on the choice of actions?

What would an optimal policy look like that would choose different actions under different states? If the optimal policy isn’t a function of states, how would you motivate the MDP formulation? Even if the optimal choices of the hyper-parameters aren’t state-dependent, do they have fixed points under some simplified setup? For example, can you describe a simple setup where an optimal number of replay iterations exists and explain what it means for a particular number of replay iterations to be optimal?

A second weakness is the lack of some crucial details. For example, as a DQN-style RL is used, it involved a replay buffer of its own. Details of the size of that replay aren’t given.

Moreover, using RL on top of CL increases the computation compared to the base CL methods with fixed hyper-parameters. Depending on how much extra computation RL is using, it can have implications on the fairness of the experimental setup. Would it be fair to compare methods with vastly different amounts of computation, especially, if the competing methods are using substantially less computation?

It will be much appreciated if you could provide some details on the experimental setup. When a task is presented, how many samples of that task are presented to the learner at a time? Are the samples presented online to the learner iid or sequential? The paper indicates that they are sequential: “The online CL setting is more challenging: … the sampling distribution is non-IID”. There could be pseudocode describing this process of interaction, not just the algorithm.

There are other works that use a validation set for continual learning such as “Large Scale Incremental Learning” by Wu et al. These should be discussed and compared with the proposed approach.

The symbols of Equation (1) aren’t described.

The x-axis labels of plots in Figure 4 aren’t given. Variable names are used for plot titles instead.


**Summary Of The Paper:**

This paper takes a particular continual learning setup and addresses the challenge of choosing two important hyper-parameters of standard continual learning approaches based on experience replay: the online-to-memory loss ratio, and the number of replay iterations. A DQN-style reinforcement learning approach is proposed to adapt these two parameters online. An additional small test memory is introduced, which is used to construct the reward and states. The efficacy of the approach is demonstrated using CIFAR datasets.


**Summary Of The Review:**

The paper addresses an important variant of continual learning, and the proposed method is simple and applicable on top of many existing approaches. On the other hand, the application of RL isn’t well-motivated, and some crucial details are missing that may warrant reevaluation of the proposed methods, such as by achieving a computational parity with competing methods.

---

> ### Author Response · Authors · 2021-11-23
> **Response to Reviewer hbHU's comments**
>
> Thanks for your constructive comments and support.
>
>
> >The main weakness of the work is the lack of clarity of the MDP formulation. In order to formulate a problem with an MDP, there need to be meaningful state-to-state transitions, and the optimal policy needs to be a function of the state. To understand whether those are present in the adopted continual learning setup, one could take a simplified scenario where the overall process of the continual learning setup is stationary. In that scenario, could you give a sensible description of the state-to-state transition dynamics that is contingent on the choice of actions?
>
> Regarding the state-to-state transition, we employ a one-step MDP (contextual bandits) which does not involve state-to-state transition. Our rationale behind this design choice is that we found the state-to-state transition involves a complex process including incoming data, sampling of replay memory, gradient update of the neural network, and the selection of action. In this process, the action selection is only playing a small part. Therefore, to simplify the problem, we employ a one-step MDP.
>
>
> >What would an optimal policy look like that would choose different actions under different states? If the optimal policy isn’t a function of states, how would you motivate the MDP formulation? Even if the optimal choices of the hyper-parameters aren’t state-dependent, do they have fixed points under some simplified setup? For example, can you describe a simple setup where an optimal number of replay iterations exists and explain what it means for a particular number of replay iterations to be optimal?
>
> About the optimal policy, we analyze the learned Q function and found the action is related to the input state space, like the test memory loss and training memory loss. Generally, a high training memory batch loss shows the sampled memory batch is probably challenging and has not been learned well by the continual learning agent and thus requires more memory iteration. However, if the test memory loss is high, which suggests the likelihood of overfitting replay memory, more memory iterations is likely sub-optimal. On the contrary, if the test memory loss is low, it is safer to employ a high memory iteration number. Therefore, we assume the optimal policy is a function of states. We will include an analysis of the learned optimal action based on a Q function and a bandit setting without a state space as a baseline in the updated version of the paper.
>
>
> >A second weakness is the lack of some crucial details. For example, as a DQN-style RL is used, it involved a replay buffer of its own. Details of the size of that replay aren’t given.
>
> Due to the page limit, we included RL implementation details including replay buffer size in appendix A2.
>
>
> >Moreover, using RL on top of CL increases the computation compared to the base CL methods with fixed hyper-parameters. Depending on how much extra computation RL is using, it can have implications on the fairness of the experimental setup. Would it be fair to compare methods with vastly different amounts of computation, especially, if the competing methods are using substantially less computation?
>
> We agree that the computation cost of RL should be discussed in the paper. We argue that our closed-loop framework supporting online hyperparameters is computationally efficient compared to offline hyperparameter tuning. It can explore a large hyperparameter space with a single run while traditional hyperparameter tuning in CL requires multiple runs of the whole data. More importantly, the multiple runs of whole data violates the fundamental assumption of continual learning: previous task data is not available.
>
>
> >It will be much appreciated if you could provide some details on the experimental setup. When a task is presented, how many samples of that task are presented to the learner at a time? Are the samples presented online to the learner iid or sequential? The paper indicates that they are sequential: “The online CL setting is more challenging: … the sampling distribution is non-IID”. There could be pseudocode describing this process of interaction, not just the algorithm.
>
> The experiment details can be found in Appendix A2. A batchsize of 10 samples is used for the continual task. The samples are iid within a task, but the data distribution is different across different tasks. Pseudocode will be included in the updated version.

---

### Official Review · Reviewer_GhFg · 2021-11-02

**Correctness:** 2
**Technical Novelty And Significance:** 3
**Empirical Novelty And Significance:** 2
**Recommendation:** 3
**Confidence:** 4

**Main Review:**

############## Strengths ##############

1. The motivation for the problem of hyper-parameter tuning in a continual learning (CL) setting is strong and appealing, and has been understudied in the literature
2. The proposed approach of RL-based search seems promising and is (for the most part) described in a good level of detail
3. The empirical evaluation goes beyond looking at performance and presents a deeper study of why and how certain aspects of the approach are working

############## Weaknesses ##############

1. The description of the approach as class-incremental appears to be incorrect
2. Some crucial details of the experimental setting are unclear in terms of the fairness of comparison against baselines
3. Some of the primary empirical results are inconsistent with the motivation throughout the paper of using RL for hyper-parameter tuning

############## Arguments ##############

The problem setting itself studied in the submission is fairly novel: how to tune the hyper-parameters in a CL setting without "cheating" and looking at future tasks while tuning the hyper-parameters. Even though this sounds trivial in hindsight, most current CL papers unfortunately tune hyper-parameters in the entire set of tasks, which is akin to training on the test set in standard single-task supervised learning. The authors then adequately motivate the use of RL-based search as a tool for hyper-parameter tuning, explaining that a replay "test" memory can be used as a proxy for CL performance. The approach itself is then clearly explained and motivated: there are pieces of information like performance on the current task and performance on previous tasks that should intuitively impact the best choice for CL hyper-parameters at the current training step. At a high level, this description is clear, insightful, and easy to follow. As a side note, I quite like the idea of using test memory as a proxy for test data. It's interesting because this actually comes from training data: the model had initially used these data points for training. So the only reason this is not training accuracy is that the model eventually would forget these samples if not for the replay strategy. Basically it's a "test-for-forgetting" memory. I wonder if models typically forget more easily test samples vs training samples (e.g., if training accuracy usually decays differently than test accuracy). I would encourage the authors to include a discussion about this in a later revision of their work, though I did not think the lack of it was a major shortcoming of the submission.

For all these positive factors, unfortunately the paper has some fundamental flaws that would need to be addressed prior to being ready for publication.

The first one is the incorrect use of the term class-incremental learning as a descriptor for their method. While the authors discuss the definition of class-incremental learning correctly, their defense for why their method using task-specific Q-functions does not break the basic assumptions of class-incremental learning is incomplete. It boils down to the difference discussed by Van de Ven et al. between two categorization schemes: single-head vs multi-head [1,2] and task- vs domain- vs class-incremental. The latter categorization, which the authors assume in this work, is _not_ about the model used for learning (e.g., whether task-specific output heads are used or not), but rather about the problem setting itself. In particular, the class-incremental setting demands that approaches are not given access to task descriptors _even during training_. And this goes against the proposed method for using task-specific Q-functions. While this may seem like a simple notation disagreement, empirically all class-incremental baselines are trained without access to task indicators, which is unfair.

Moreover, on the point of task-specific Q-functions, it remains unclear exactly _why_ the authors chose to train task-specific Q-networks. It is also unclear exactly _how_ this is done: is the replay buffer for Q learning cleared upon the start of each new task? If so, why? One would think intuitively that, if the task-agnostic state space defined in Sec 4.2 and 4.3 indeed captures a relation between test-memory performance and hyper-parameters, then this should be roughly agnostic to the task. It is unclear in the paper why the authors chose not to. Back to the previous point, if this choice is indeed ideal, then other methods should also have access to task indicators.

Another crucial detail that is somewhat unclear in the paper is the relation between replay memory size and test memory size. In particular, the combined sizes of these two memories should be equal to other methods' total replay memory. However, some of the wording in Sec 3 seems to suggest that this is not the case, and the test memory used for RL training uses _additional_ memory that the base methods are not allowed to use. In order to properly assess the benefits of the proposed approach, it should be evaluated under the same conditions as other approaches, concretely in terms of storing the exact same number of samples in memory. The evaluation should gauge whether increasing the replay memory size is better than using the additional memory as a test memory to follow the RL approach.

One more choice remains unclear: why only test the benefits of RL-based search for individual hyper-parameters? Would the method not be applicable to searching over multiple hyper-parameters simultaneously (e.g., by training multiple RL agents in parallel)? And somewhat related: why the choice of using discrete actions? While it seems valid, since it is akin to doing a grid-search over a choice of possible hyper-parameters, it is unclear why not just use a continuous action space which RL supports.

It is also never stated what the exploration strategy is for DQN. I assume epsilon-greedy? This choice seems to be quite relevant for the problem at hand, because the RL agent is being evaluated from the start (i.e., if it is bad initially, that will degrade the overall performance of the CL agent, as it will likely never recover performance on the earlier tasks). It seems like the algorithm may only work because all the choices of hyper-parameters are reasonably good. What if we don't know a priori a good way to restrict the space of possible hyper-parameters? This is for example very hard if we apply the proposed ideas to something like EWC, where the possible range of \lambda is huge. Even in the case of replay methods, it's unclear what strategy the authors used to restrict the action space to the selected values.

In terms of results, unfortunately the findings do not seem to entirely support the claims made throughout the paper, and in particular the motivation itself for using RL. Concretely, the trend of hyper-parameter adaptation in Figure 4 suggests that the optimal hyper-parameters are _fixed_ and not state-based. Therefore, there doesn't seem to be a point at all in using RL, since the actions should not be state-dependent. This conclusion is further validated by the fact that ER and SR with a fixed value of 3 replay iterations outperform the RL-based method. This, along with the fact that the authors in the end use a single-step MDP, suggests that perhaps a multi-armed bandit formulation would have been more appropriate.

############## Additional feedback ##############

The following points are provided as feedback to hopefully help better shape the submitted manuscript, but did not impact my recommendation in a major way.

Intro
- The first paragraph is a bit disorganized
- Connection to other works for online hyper-parameter tuning seems weak: only downside pointed is that there is no held-out validation data. It becomes clear only later that this validation data comes from _other_ tasks beyond those in the CL problem, which is not always available.
- I wonder how specific the approach is to replay. Could it be used directly for other techniques, e.g. regularization-based?
- The intro overall is quite complete, albeit a bit disorganized. The examples in the first paragraph seem to come out of nowhere, the method is introduced twice seemingly independently in the penultimate and last paragraph.

Sec 2
- Overall the connections in the CL section are drawn nicely
- The explanation here of using "external" validation data explains why this is stated as a weakness in the Intro: external means from tasks _not_ in the sequence, which may or may not require different hyper-parameters (and there may not be access to such external tasks)
- The connection to RL-based hyper-parameter tuning strategies is also nice

Sec 4.1
- The definition of replay ratio is quite confusing. It's not really a step size, and a ratio suggests a ratio of "the amount of replay vs new data". From Eq. 1, it seems that it's instead a weighting of new vs replayed data.

Sec 4.2
- The state space is carefully designed and the intuition behind it is clearly explained.
- The choice of doing one-step MDP is buried at the end of the section. I think this choice is valid, because it might not be necessary to tune hyper-parameters for future learning. However, the use of an RL formulation throughout the paper suggests that this is exactly what this work is doing. I suggest placing this caveat much earlier in the paper and discuss the implications of this choice.

Sec 4.3
- Overall 4.2 and 4.3 are well written and it's nice that the "default" action is explained. One comment is that the state space for 4.3 is not described in nearly as much detail as in 4.2. In particular, it's not explained why the state spaces are different.


Sec 5
- Task sequences are fairly short, but I don't think that matters much since replay has been shown to work well on long sequences and I see no technical reason why this approach would be more susceptible to the sequence length
- Figure 4 needs x-axis label (I assume # task?)

Appendices
- Many appendices are never mentioned in the main paper, including the forgetting curves of Fig 5

[1] Farquhar and Gal. Towards robust evaluations of continual learning. arXiv:1805.09733, 2018.

[2] Chaudhry et al. Riemannian walk for incremental learning: Understanding forgetting and intransigence. arXiv:1801.10112, 2018.

**Summary Of The Paper:**

This submission proposes an RL-based method for tuning hyper-parameters of continual learning (CL) methods. The rationale is that tuning hyper-parameters on the entire set of tasks off-line invalidates the CL assumption of online learning, and therefore online methods are needed. The authors discuss the elements that make up the state space for making hyper-parameter choices, such as various performance metrics, and use use (deep) Q-learning to train an RL agent to make decisions about the hyper-parameter to use at each training step. The approach is applied on top of two existing replay-based methods, varying two hyper-parameters (one each in separate evaluations), and demonstrated to achieve improved performance w.r.t. the _un-tuned_ variants.


**Summary Of The Review:**

Unfortunately, I recommend the rejection of this paper. I believe that the problem that the authors are proposing is insightful in itself, and certainly captures one current issue with most CL evaluations today: that most approaches tune hyper-parameters in an off-line setting, which is inconsistent with the CL problem formulation. However, in my opinion there are a number of shortcomings (primarily empirical) that should be addressed for turning this into a complete conference publication. Most critically, there are a number of issues in the evaluation setting and in the results themselves that should be addressed prior to publishing the work. I do strongly encourage the authors to continue improving upon their work, as I believe that it could be highly impactful.

---

> ### Author Response · Authors · 2021-11-23
> **Response to Reviewer GhFg's comments**
>
> Thanks for your detailed comments and valuable feedback.
>
>
> > I wonder if models typically forget more easily test samples vs training samples
>
> In the revised paper, we include using training memory loss as a baseline for approximation of test loss. As shown in the updated Fig 2,  the use of train loss leads to a poor approximation of test loss, especially when more continual learning tasks are encountered, due to the overfitting of the replay memory.
>
>
>
> > the class-incremental setting demands that approaches are not given access to task descriptors even during training. And this goes against the proposed method for using task-specific Q-functions. While this may seem like a simple notation disagreement, empirically all class-incremental baselines are trained without access to task indicators, which is unfair.
>
> We would like to clarify that the training of a task-specific Q-function does not use an explicit task indicator. This method can be easily adapted to expand the Q network every time a new class is encountered, eliminating the need for a task indicator.  In this way, a task indicator is not used. We will make this description clearer in the updated version of the paper.
>
>
> >Moreover, on the point of task-specific Q-functions, it remains unclear exactly why the authors chose to train task-specific Q-networks. It is also unclear exactly how this is done: is the replay buffer for Q learning cleared upon the start of each new task? If so, why?
>
> The rationale behind the task-specific Q-function design is that some continual learning tasks may be similar to previous tasks and some may be quite different from previous ones. The state-space design is mainly focused on the model's performance on the current tasks and previous tasks. There may be some other characteristics regarding the relationship of tasks that have not been captured by the losses. Therefore, we think the state-action function may vary from task to task.
>
>
> >Another crucial detail that is somewhat unclear in the paper is the relation between replay memory size and test memory size.
>
> We agree that different approaches should store the same number of samples to make a fair comparison. This will be clarified in the next version of the paper.
>
>
>
> >One more choice remains unclear: why only test the benefits of RL-based search for individual hyper-parameters? Would the method not be applicable to searching over multiple hyper-parameters simultaneously (e.g., by training multiple RL agents in parallel)? And somewhat related: why the choice of using discrete actions? While it seems valid, since it is akin to doing a grid-search over a choice of possible hyper-parameters, it is unclear why not just use a continuous action space which RL supports.
>
> The challenge to learn multiple hyper-parameters simultaneously mainly lies in the sample efficiency. With online learning and a single CL agent, there are only a few thousand samples collected in the continual learning dataset. We will explore continuous control of the multi-dimensional hyper-parameter space in future work
>
>
> >It is also never stated what the exploration strategy is for DQN. I assume epsilon-greedy? ... It seems like the algorithm may only work because all the choices of hyper-parameters are reasonably good. What if we don't know a priori a good way to restrict the space of possible hyper-parameters? ...
>
> Due to the page limit, we put the implementation details of RL in the appendix (A2). $\epsilon$-greedy is used for exploration with a decaying epsilon. For the replay buffer of RL, we use the most recent 100 samples. We agree that the action space indeed influences the results as RL randomly explores the action space at the early stage of continual learning problems (e.g. the first two tasks). However, we found the RL is relatively robust to the choice of action space in the replay-based method since for all the three datasets and two backbones algorithms (ER and SCR) we have been using the same action space.
>
>
>
> >the trend of hyper-parameter adaptation in Figure 4 suggests that the optimal hyper-parameters are fixed and not state-based. Therefore, there doesn't seem to be a point at all in using RL, since the actions should not be state-dependent.
>
> The results in Fig 4 only focus on the action distribution changes across different tasks and lose the information about action distribution change within a task. In the revised paper, we update Fig 4 to show what is the greedy action selected by the RL agent in each optimization step. This figure shows that the action is not fixed on a single value. For example, in the case of ratio adaptation, although the agent is mostly focused on smaller values of the incoming ratio, large values are still considered to be the greedy action by the RL agent once in while. The selection of a large incoming ratio once in a while is likely beneficial to combat outfitting on replay memory.

---

> > ### Comment · Reviewer_GhFg · 2021-11-23
> > **Continued discussion**
> >
> > I thank the authors for their response. I unfortunately believe that it doesn't address the fundamental issues with this submission, and therefore will not increase my score. I reply below to individual comments by the authors.
> >
> > - **Train vs. test memory**. While I appreciate that the authors included the additional comparison to training memory, unfortunately not enough details are given to understand the new claims. Is "train memory" in Fig. 3 the same as the "replay memory" from the rest of the paper? If so, this is not really what I was suggesting, and I frankly don't find that this evaluation adds more value. If not, please clarify exactly what the difference is between "train memory" and "test memory". My suggestion was that the "test memory" used to compute the RL loss is in fact a collection of training samples of earlier tasks (so, in a way, it already is a training memory). Intuitively, using this "test memory" is useful even though its constructed from training samples because the agent stops training on it, and so performance on the "test memory" is affected by forgetting. However, it is possible that forgetting affects training data (as used for constructing the "test memory") at a different rate than it affects _true_ test data (as used to evaluate the true test loss). My recommendation was to test whether this is actually the case; if forgetting is similar across training data (e.g., from the "test memory") and test data (from the real test set), then that validates the choice to construct the "test memory" from training samples. I'd like to emphasize that this was merely a suggestion and it does not play a major role in my rating of this submission.
> > - **Class-incremental learning**. Training separate Q-functions requires as much access to a task indicator as would training separate classifiers for each task. If the former is allowed, then the latter should be allowed (especially for baselines) as well.
> > - **Task-specific Q-functions**. This is a nice description of the rationale, and should be included in the paper as it gives a clearer intuition for why the use of separate Q-functions might be useful. The authors still failed to explain _how_ this is done, though: is the replay buffer filled with data from previous tasks or only data from the current task?
> > - **Multiple hyper-parameters and continuous search spaces**. While I agree that perhaps tuning multiple hyper-parameters might not be feasible given the few samples collected during RL training, this seems to substantially undermine the usefulness of the approach. If it is not possible to tune more than a single hyper-parameter, how can one choose which hyper-parameter to tune? How should one choose the remaining hyper-parameters?
> > - **Exploration strategy**. Thank you for the clarification of the search strategy used for RL exploration. I disagree with the statement that "RL is relatively robust to the choice of action space", precisely because only one choice of hyper-parameters was used. When doing off-line hyper-parameter tuning, there is no harm to trying poor hyper-parameters, because they don't affect the performance of the agent in any way. In this formulation of online tuning, that is not the case: if the agent has a choice of a very poor hyper-parameter value, that may be catastrophically damaging to the CL process. It would be ideal if the hyper-parameter tuning strategy were robust to poor hyper-parameter values, as in the off-line case.
> > - **Fixed action selection**. The new figure doesn't really show substantial differences with the original one in terms of showing that the agent converges to an almost fixed action selection strategy. The authors claim that "the selection of a large incoming ratio once in a while is likely beneficial to combat outfitting on replay memory." By this same logic, one could infer that the selection of a low number of iterations once in a while is likely beneficial as well. This is explicitly shown to _not_ be the case in the bottom rows of Table 2, where it becomes apparent that a fixed strategy of always selecting the highest number of iterations is better. Therefore, I am not convinced by the authors claim regarding the incoming ratio.

---

### Official Review · Reviewer_vSnQ · 2021-11-02

**Correctness:** 3
**Technical Novelty And Significance:** 2
**Empirical Novelty And Significance:** 3
**Recommendation:** 5
**Confidence:** 4

**Main Review:**

Choosing appropriate hyper-parameters for replay differently for every task, and doing this using closed-loop control is an interesting idea.

But it's not clear to me if the improvement in performance is worth all the extra machinery required to train a DQN to choose the hyper-parameters. Moreover, from the results of Fig. 4, it seems like the DQN network seems to go towards a fixed action rather than adaptively change the hyper-parameters in a complex way. This indicates that there might be a way to choose these hyper-parameters using a much simpler setup.

The authors do not discuss the computational cost-performance tradeoff of using a DQN to choose the hyper-parameters, vs using a simpler open-loop scheme to choose them. This is the key issue that requires addressing.

The paper is generally clearly written, and their algorithm itself is relatively clearly explained.
But the authors go through a big chunk of related work, including some which are not of immediate relevance to the algorithm in the paper. This section can easily be shortened.

The empirical evaluation of their method covers quite a lot of ground, and makes a convincing case that their method does have some advantage in terms of performance.

A couple of minor comments:
- Fig 3 is tiny and the plot is really hard to read without zooming in a lot.
- Fig 4: x-axis is not labelled, and not clear what it is.

**Summary Of The Paper:**

In this paper, the authors use reinforcement learning (specifically Q-learning) to choose certain hyper-parameters of replay for online continual learning. These hyper-parameters are chosen online using a DQN network as it sees new data, continuously adapting the choice of hyper-parameters based on a separate test memory.

**Summary Of The Review:**

Overall, the significance and novelty of the work is not so clear to me, since it seems like it's a straightforward application of an off-the-shelf RL algorithm for choosing parameters of online continual learning. On the other hand, it is interesting that this setup does give performance improvements. But the biggest reason for my score is that the tradeoff between adding the complexity of DQN for choosing hyper-parameters vs performance gains is not discussed or quantified.

---

> ### Author Response · Authors · 2021-11-23
> **Response to Reviewer vSnQ's comment**
>
> Thanks for your constructive comments and feedback.
>
>
> >Moreover, from the results of Fig. 4, it seems like the DQN network seems to go towards a fixed action rather than adaptively change the hyper-parameters in a complex way.
>
> The results in Figure 4 only focus on the action distribution changes across different tasks and lose the information about action distribution change within a task. In the revised paper, we update Figure 4 to show what is the greedy action selected by the RL agent in each optimization step. This figure shows that the action is not fixed on a single value. For example, in the case of ratio adaptation, although the agent is mostly focused on smaller values of the incoming ratio, large values are still considered to be the greedy action by the RL agent once in while. The selection of a large incoming ratio once in a while is likely beneficial to combat outfitting on replay memory.
>
>
> >The authors do not discuss the computational cost-performance tradeoff of using a DQN to choose the hyper-parameters, vs using a simpler open-loop scheme to choose them. This is the key issue that requires addressing.
>
> Regarding the computational cost, the RL-based online hyperparameter tuning is especially useful to explore a large hyper-parameter space. Compared to offline hyperparameter tuning, the proposed method can explore all the hyperparameters in a single run, while offline hyperparameter tuning requires repetitive runs of the whole data.

---

> > ### Comment · Reviewer_vSnQ · 2021-11-26
> > **Concerns not adequately addressed**
> >
> > I thank the authors for their response.
> > Unfortunately my concerns are not addressed in the author response.
> > I am looking for something a lot more quantitative and significant on the cost-performance tradeoff, since I think that's a fairly important issue for this paper.
> > Therefore, my score will remain the same.

---

### Official Review · Reviewer_ng9G · 2021-11-03

**Correctness:** 4
**Technical Novelty And Significance:** 2
**Empirical Novelty And Significance:** 3
**Recommendation:** 5
**Confidence:** 4

**Main Review:**

On the positive side, the approach presented for the control of the parameters is well described.  The datasets used and the selection of baselines is in line with the state of the art in this problem area.  And the quality of presentation is quite good with few noticeable errors or confusing passages.

On the other hand, the presented approach is built off of ER or SCR as the backbone and using DQN to perform the required reinforcement learning.  So the theoretical contribution is mostly centered in the formulation of the reward signal which is then solved in a known way.  The reward function under test, in this case, is also not particularly sophisticated either, boiling down to measuring the difference in total risk from one time step to another on a random holdout set.  Alternatively, other additional ways to encapsulate the risk might have been good to test to characterize how this RL technique is affected by these choices or bringing in a more elaborate method of choosing the replay set would have been possible.  Figure 4 is somewhat confusing matters also in that it appears to almost contradict the non-stationary nature that we are trying to capture, as it appears that at least for this example that over time the parameters mostly converge towards a small region.  Finally, unless I am mistaken in my reading of Table 2, but it appears that on cifar when multiple iterations are used this method does no better than plain SCR and shows almost no difference with the ER method if multiple iterations are allowed.


**Summary Of The Paper:**

The authors present an approach for continual learning based on replay.  By finding a way to represent hyperparameters of the replay learning process, specifically the replay ratio and the amount of replay iterations, the authors use a reinforcement learning approach to attempt to find the optimal parameters for each epoch.  The impetus for this design is ascribed to the fact that in the continual learning regime it is expected that the input distribution is not stationary and so we would also like to be able to modify the parameters automatically in response to the change in present data.  The authors provide experimental results for this method on cifar10, cifar100, and CLRS and by using their approach in complement to Experience Replay (ER) and Supervised Contrastive Replay (SCR).

**Summary Of The Review:**

While the paper was well written and the ideas made sense as presented, however, my concerns with the amount of theoretical novelty combined with an empirical situation in which I have some questions is going to make my recommendation at 5, marginally below threshold.

---

> ### Author Response · Authors · 2021-11-23
> **Response to Reviewer ng9G's comments**
>
> Thank you for your constructive feedback. In the following, we address it point by point.
>
>
> > The reward function under test, in this case, is also not particularly sophisticated either, boiling down to measuring the difference in total risk from one time step to another on a random holdout set. Alternatively, other additional ways to encapsulate the risk might have been good to test to characterize how this RL technique is affected by these choices or bringing in a more elaborate method of choosing the replay set would have been possible
>
> About the reward design, although it is not a complex method, we believe it is novel and effective in the sense that it is the first method to identify an online feedback signal for hyperparameter tuning. Previous continual learning methods have been using training loss or external validation data for hyper-parameter tuning. The use of external validation data violates the fundamental assumption of a continual learning setting. In the revised paper, we include using training memory loss as a baseline for approximation of test loss. As shown in the updated Figure 2 it turns out the use of train loss leads to a poor approximation of test loss due to the overfitting of the replay memory. This result suggests the additional test memory design is necessary for obtaining reliable feedback signal on model's performance.
>
>
> >Figure 4 is somewhat confusing matters also in that it appears to almost contradict the non-stationary nature that we are trying to capture, as it appears that at least for this example that over time the parameters mostly converge towards a small region
>
> We agree that Figure 4 is not the best way to present the learned action selection scheme, as it only focuses on the action distribution changes across different tasks and loses the information about action distribution change within a task. In the revised paper, we update Figure 4 to show what is the greedy action selected by the RL agent in each optimization step. This figure shows that the action is not fixed on a single value. For example, in the case of ratio adaptation, although the agent is mostly focused on smaller values of incoming ratio, large values are still considered to be the greedy action by the RL agent once in while. The selection of a large incoming ratio once in a while is likely beneficial to combat outfitting on replay memory.
>
>
>
> >Finally, unless I am mistaken in my reading of Table 2, but it appears that on cifar when multiple iterations are used this method does no better than plain SCR and shows almost no difference with the ER method if multiple iterations are allowed.
>
> The number of memory iteration is regarded as a hyperparameter. We agree that the proposed method's performance does not always outperform the results with the optimal hyperparameter. However, in practice, the optimal hyperparameter is unknown and difficult to identify in continual learning. The aim of the proposed method is to try to find suitable hyperparameter settings in an online manner with a single run of data.

---

### Decision · Program_Chairs · 2022-01-20

**Decision:**

Reject

**Comment:**

This manuscript studies the problem of continual learning and introduces a reinforcement learning agent to select hyperparameters for replay/training. Ordinarily, replay based mechanisms for continual learning use settings and hyperparameters that are chosen and fixed through training. If it was possible to adjust replay dynamics online (in this case by looking at performance on a held-aside test set), performance might be improved. This is the approach taken by this manuscript.
Reviewers were generally happy with the writing of the paper and presentation of the material. At the same time, more than one reviewer worried about the novelty of the approach. In essence, the proposal amounts to using a black-box optimizer (in this case RL) to adjust online the hyperparameters (e.g. the replay ratio) for continual learning (of-the-shelf ER and SCR). Viewed through this lens, and given that the optimizer in this case was a straightforward application of DQN, this concern is potentially well founded. The primary novelty then is the construction of the reward function to be optimized: in this case defined as the decrease of the CL loss measured on a held aside test set that is constructed online. Nevertheless, novelty is only part of the equation and strong empirical results can easily be a deciding factor in readiness for publication. On this front, reviewer GhFg points out that the empirical results and comparisons with baseline methods are not as clear as they need to be. Several issues are raise in discussion: the primary one is around the question of how the authors have allowed task-specific information for the Q functions used by RL, and what the implications of this might be. The baselines compared against do not use any task-specific information, which muddies the waters when trying to understand the comparisons. I agree with the reviewer that the manuscript needs to do a better job of making the empirical setting and comparisons as transparent and fair as possible. Given this, and the fact (raised by several reviewers) that some empirical evidence presented in the manuscript actually points to RL selecting near-static parameters over time, I recommend that the manuscript be rejected. At the same time, I want to encourage the authors to focus on a streamlined version of the manuscript that addresses the issues raised by GhFg, as I believe that if the concerns can be addressed the work is close to making a compelling contribution for the field.